evolution

Allen's rule, climate adaptation, environmental extremes, thermoregulation

**Author for correspondence:**
Katie LaBarbera
e-mail: klabarbe@umn.edu

†Present address: Dept. of Ecology, Evolution, and Behavior, College of Biological Sciences, Ecology Bldg, University of Minnesota – Twin Cities, Falcon Heights MN 55108.
‡Present address: San Diego Zoo Institute for Conservation Research, 15600 San Pasqual Valley Rd, Escondido CA 92027, USA.

# Context-dependent effects of relative temperature extremes on bill morphology in a songbird

Katie LaBarbera[1,†], Kyle J. Marsh[2], Kia R. R. Hayes[1] and Talisin T. Hammond[1,‡]

[1]Museum of Vertebrate Zoology, Department of Integrative Biology, University of California – Berkeley, Berkeley CA 94720, USA
[2]Point Blue Conservation Science, 3820 Cypress Drive, Ste #11, Petaluma, CA 94954, USA

 KL, 0000-0002-2000-7280

Species increasingly face environmental extremes. Morphological responses to changes in average environmental conditions are well documented, but responses to environmental extremes remain poorly understood. We used museum specimens to investigate relationships between a thermoregulatory morphological trait, bird bill surface area (SA) and a measure of short-term relative temperature extremity (RTE), which quantifies the degree that temperature maxima or minima diverge from the 5-year norm. Using a widespread, generalist species, *Junco hyemalis*, we found that SA exhibited different patterns of association with RTE depending on the overall temperature regime and on precipitation. While thermoregulatory function predicts larger SA at higher RTE, we found this only when the RTE existed in an environmental context that opposed it: atypically cold minimum temperature in a warm climate, or atypically warm maximum temperature in a cool climate. When environmental context amplified the RTE, we found a negative relationship between SA and RTE. We also found that the strength of associations between SA and RTE increased with precipitation. Our results suggest that trait responses to environmental variation may qualitatively differ depending on the overall environmental context, and that environmental change that extremifies already-extreme environments may produce responses that cannot be predicted from observations in less-extreme contexts.

## 1. Introduction

As climate change is predicted to increase the extremity and variability of environmental conditions [1,2], understanding the

effects of environmental extremes on phenotypes is a priority [3]. Climate extremes can have large impacts on ecosystem function and organismal fitness: recent extreme events have caused mass mortality events [4–7], reproductive failure [8] and shifts in community structure [9,10]. Understanding responses to environmental extremes, particularly how they differ from linear extensions of responses to less-extreme conditions, will be crucial in predicting and understanding the changes to come [2,3,11].

Characterizing the evolutionary effects of environmental extremes is challenging due to the rarity and inherent unpredictability of extreme events [3]. Natural history collections with deep sampling over time can resolve this difficulty by capturing historical responses to past environmental conditions, including extreme conditions [12]. The avian bill is particularly well-suited to such examination of historical responses due to its preservation in museum specimens [13] and its strong ties to fitness (e.g. [14,15]).

The avian bill is shaped by dietary [15], communicative [14] and thermoregulatory [16,17] demands. Bill morphology can affect thermoregulation over several time scales: it can evolve over generations [18]; shrink or grow plastically over weeks in response to external temperatures [19–21]; or have its heat loss modulated within minutes through vasodilation or vasoconstriction of blood vessels within the bill [17,22]. The relationship of bill surface area to environmental conditions has been relatively well studied over decade-plus time scales, with bill surface area being linked to maximum temperatures in summer [20,23], minimum temperatures in winter [24,25] and habitat type [13,26,27]. Following Allen's Rule, which predicts that morphological extremities will be smaller in cold environments [24,28], thermoregulatory theory predicts that bill surface area will be larger in warmer environments and smaller in cooler environments [24], a phenomenon supported by empirical evidence [17,27,28]. Thermoregulatory theory and empirical evidence further suggest that this relationship will be stronger in higher humidity, because humidity intensifies avian dependence on non-evaporative cooling mechanisms [29].

Responses of bill surface area to climate at shorter time scales, which could capture brief but extreme environmental fluctuations, remain comparatively unexamined. Here, we develop a measure of relative temperature that quantifies the extremity of temperature within the context of a location's usual thermal regime. This relative measure accounts for the possibility that the impact of environmental variation on an organism may depend not only upon the absolute value of that variation, but also upon its value in comparison to the baseline conditions usually experienced by that organism [30]. Any measure of extremes contains an implicit baseline in contrast with which it is extreme; this relative measure makes that baseline explicit, and in varying it according to local norms, renders this measure broadly comparable across climates. By analysing this measure in conjunction with absolute temperature and precipitation, it becomes possible to tease apart how interactions between absolute environmental context and relative extremity of conditions influence bill morphology. Using a high-resolution, museum collection-based dataset, we characterized the short-term (one year) effects of relative temperature extremity in summer on bill surface area in dark-eyed juncos (*Junco hyemalis*) in California, USA, over a 50-year time period. We predicted that, in accordance with thermoregulatory pressures as posited by Allen's Rule, bill surface area would be positively related to short-term relative temperature extremity (relative minimum and maximum temperatures). We further predicted that these relationships would be modulated by environmental context in the form of the absolute temperature regime and precipitation (as a proxy for humidity), expecting that the relationship with relative minimum temperature would be stronger at colder absolute temperatures, the relationship with relative maximum temperature stronger at warmer absolute temperatures, and both relationships stronger at higher levels of precipitation.

# 2. Material and methods

## 2.1. Study species

Dark-eyed juncos are medium-sized (approx. 16 g) songbirds common across most of North America. They subsist on seeds, arthropods and occasionally fruit [31]. Because their breeding season (spring through early fall) home ranges are small (mean ≤200 m$^2$; [32]), a junco captured during the breeding season can reasonably be assumed to have spent the season near its capture location. The junco's metabolic rate increases as external temperatures decrease from 22.5°C to −10°C [33]. That juncos increasingly prioritize heat conservation over reducing predation risk as temperatures decrease [34] demonstrates the value of thermoregulation to these birds.

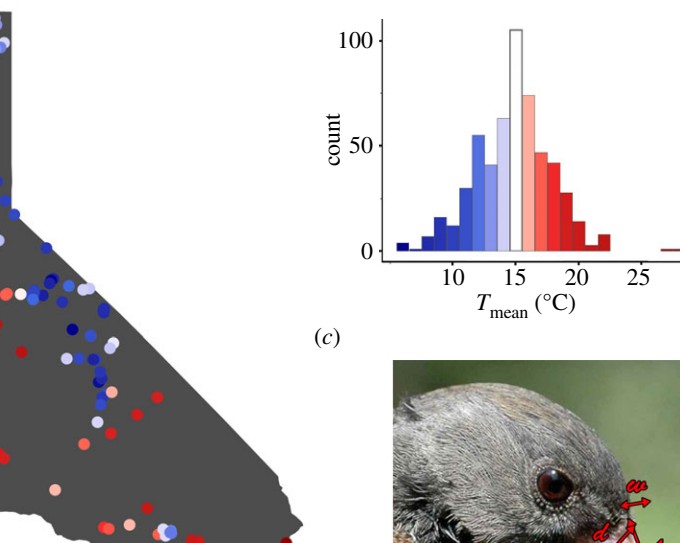

**Figure 1.** Specimen collection locations across California (*a*), coloured according to the 5-year mean temperature. The associated temperatures for each colour are given in (*b*), which also shows sample size across temperatures. The bill measures from which surface area was calculated, length *l*, depth *d* and width *w*, are shown on the study species *Junco hyemalis* (*c*).

## 2.2. Morphological data collection

We measured museum specimens of two subspecies of dark-eyed junco belonging to the 'Oregon junco' group, *Junco hyemalis pinosus* and *J. h. thurberi*, collected between 15 March and 30 September from 1900 to 1950 within the state of California (figure 1). We chose this 50-year timespan because it includes greater than 90% of California junco specimens in the collections we accessed; after 1950, specimens are temporally and geographically sparse. We focused on the time period between 15 March and 30 September to ensure that the individuals studied were present to breed; juncos present in the winter may be long-distance migrants wintering in California. Specimens are held in the collections of the Museum of Vertebrate Zoology or the California Academy of Sciences. Metadata associated with the specimens were downloaded from VertNet (http://vertnet.org).

Bill length, width and depth (figure 1), as well as tarsus length and wing chord, were measured by KL with digital calipers, and bill surface area was calculated from the three linear bill measurements following [26]. Further details can be found in [13].

## 2.3. Climate variables

We obtained records for four monthly climate variables from the PRISM historical climate dataset [35]: precipitation and mean, minimum, and maximum temperature. For each calendar month from April through August during the 5 years prior to the collection date, we calculated mean values for a circle with a radius of 15 km centred on the collection locality for each specimen (details in supplementary information). These values were then used to calculate the climate variables and relative extremity measures used for analysis, described below. We restricted our analysis to this specific monthly range because juncos breeding in our study area often engage in winter migrations that may take them sufficiently far to experience different climatic conditions, particularly as they often involve changes in elevation [31]. Therefore, we can be certain that we are measuring conditions experienced by the birds only when they reside on the breeding grounds, in April through August.

We calculated mean temperature and recent precipitation, as well as measures of relative extremity (see below). We calculated the measures of *mean temperature* and *temperature standard deviation* as the mean and standard deviation, respectively, of the monthly mean temperatures (in °C) for the summer months (April through August) over the previous 5 years. *Recent precipitation* was the mean monthly precipitation during the summer of the year immediately prior to collection. We included recent precipitation, rather than a longer term measure, because we expected humidity to affect the impact of relative temperature extremes on bill surface area in the moment, through reducing the immediate effectiveness of evaporative heat dissipation [29], rather than as a longer term regime.

We used a 5-year time period, here and for calculation of relative measures (below), because we wanted to capture the environment over the last several generations and no more. Five years should encompass two to three generations of juncos [31], sufficient to summarize not only the lived experience of the focal specimen but also that of its parents. This is similar to the time period used in another study of the effects of climate on bill morphology [29]. Longer time periods, such as the 30-year period used in [27], would be less appropriate for this study because our goal is to capture the effects of short-term rather than long-term variability.

## 2.4. Measures of relative temperature extremity

In this study, 'extreme' temperatures are defined in terms of comparison to an established, non-extreme baseline. This point of comparison must consist of both an absolute value and a measure of variability, and the extreme value must differ from the absolute value by an amount that exceeds that measure of 'usual' variability. For example, a temperature of 0°C in a location with a mean temperature of 10°C would be more extreme if temperatures regularly ranged from 8 to 12°C than it would be if temperatures regularly ranged from 0 to 20°C. We, therefore, calculated our measures of relative temperature extremity by comparison with both a measure of absolute temperatures over the past 5 years and a measure of temperature variability over that same time period.

*Relative minimum temperature* was calculated for each specimen by taking the difference between the mean monthly minimum summer temperature during the year immediately prior to collection and the mean of mean monthly minimum summer temperatures over the preceding 5 years, and dividing that difference by the standard deviation of mean monthly minimum summer temperatures over the preceding five years (Formula 1). *Relative maximum temperature* was calculated likewise, using maximum instead of minimum temperature values.

$$\text{Relative minimum temperature} = \left( T_{\min_1} - \frac{\sum_{i=1}^{5} T_{\min_i}}{5} \right) / \sigma_{5yrT_{\min}}.$$

**Formula 1.** Formula for the measure of relative minimum temperature. $T_{\min_i}$ is the mean monthly minimum temperature during the summer $i$ years prior to specimen collection. $\sigma_{5yrT_{\min}}$ is the standard deviation of monthly mean minimum summer temperatures during the 5 years prior to specimen collection. Relative maximum temperature was calculated in the same manner.

Greater absolute values of these relative temperature measures indicate more extreme deviations than usual (i.e. greater in proportion to the variability over the past 5 years) from the temperatures experienced over the past 5 years. The absolute temperature is not reflected by these measures, rendering them comparable across various climates: regardless of the absolute temperature regime, a large positive extremity measure indicates that the climate variable has been unusually high, while a large negative extremity measure indicates that the climate variable has been unusually low. To incorporate effects of absolute temperature into our analysis, we included the interactions of relative extremes with mean temperature, a measure of absolute temperature, in the statistical model (see below).

## 2.5. Statistical analysis

Using the gamm4 package [36] in R v. 3.4.4 [37], we ran a generalized additive mixed model fit by restricted maximum likelihood (REML) with bill surface area as the response, a nonlinear smooth term of latitude and longitude to control for spatial autocorrelation, the year of collection as a random effect to account for temporal patterns, and the following as fixed effects: mean temperature; temperature standard deviation over 5 years; precipitation for the previous year; relative mean maximum and minimum temperature for the previous year, and their interactions with mean temperature and with precipitation for the previous year; month of collection; subspecies; tarsus length, to control for effects of overall body size; and sex. This

**Table 1.** Bill surface area was influenced by temperature, precipitation, subspecies and sex. Relative temperature minimum and maximum interacted with mean temperature and with precipitation. Asterisks indicate statistical significance. $N = 516$ individuals.

| variable | estimate ± SE | t or [1]F | P |
|---|---|---|---|
| relative min temp | −5.22 ± 1.90 | −2.76 | 0.006* |
| relative max temp | 3.80 ± 2.57 | 1.48 | 0.141 |
| mean temp | 0.02 ± 0.12 | 0.21 | 0.834 |
| relative min temp × mean temp | 1.57 ± 0.52 | 3.02 | 0.003* |
| relative max temp × mean temp | −1.25 ± 0.57 | −2.19 | 0.029* |
| temp std dev | 0.27 ± 0.42 | 0.63 | 0.532 |
| recent precip | −0.02 ± 0.02 | −1.02 | 0.307 |
| relative min temp × recent precip | 0.20 ± 0.07 | 3.01 | 0.003* |
| relative max temp × recent precip | −0.17 ± 0.08 | −2.12 | 0.034* |
| month | −0.04 ± 0.17 | −0.25 | 0.807 |
| subspecies (*J. h. thurberi*) | −3.06 ± 1.02 | −3.02 | 0.003* |
| tarsus length | 1.58 ± 0.44 | 3.55 | <0.001* |
| sex (male) | 1.59 ± 0.50 | 3.12 | 0.001* |
| s(latitude, longitude) | NA | 3.17[1] | 0.007* |

model is essentially identical to running a generalized linear mixed model except that it also controls for potentially nonlinear patterns of spatial autocorrelation with the smooth term. We calculated marginal and conditional $R^2$ following [38]. For additional methodological details, including testing for multicollinearity, an analysis using wing chord as an alternative proxy for body size, and testing for Bergmann's Rule, see electronic supplementary material.

# 3. Results

## 3.1. Characterizing the statistical model

The smooth term of latitude and longitude was significant (table 1) with an estimated degrees of freedom of 5.22, indicating considerable nonlinearity in the effects of geographic location and confirming that an additive model approach was appropriate to account for spatial autocorrelation. A Moran's I test of spatial autocorrelation in the model residuals was nonsignificant ($p = 0.133$), confirming that the smooth term adequately controlled for spatial patterns. The random effect of year had a variance ± s.d. of $0.713 ± 0.844$. Residual variance ± s.d. was $28.972 ± 5.383$. Marginal $R^2$, the proportion of variance explained by fixed effects alone, was 0.117. Conditional $R^2$, the proportion of variance explained by fixed and random effects (i.e. by the entire model), was 0.398.

## 3.2. Environmental effects

Relative minimum temperature interacted with mean temperature, being negatively associated with bill surface area at low mean temperatures but reversing to a positive relationship with bill surface area at high mean temperatures (figure 2). Relative maximum temperature interacted with mean temperature in the opposite manner, with a positive effect at low mean temperatures that reversed to a negative relationship at high mean temperatures. Relative minimum and maximum temperatures also interacted with recent precipitation, with the magnitude of the relationship between relative temperature and bill surface area increasing at higher levels of precipitation (figure 3). Temperature standard deviation had no effect on bill surface area.

# 4. Discussion

The avian bill plays a substantial role in thermoregulation [17]. Therefore, we predicted that bill surface area would be positively associated with relative maximum and minimum temperature [24], and that

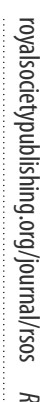

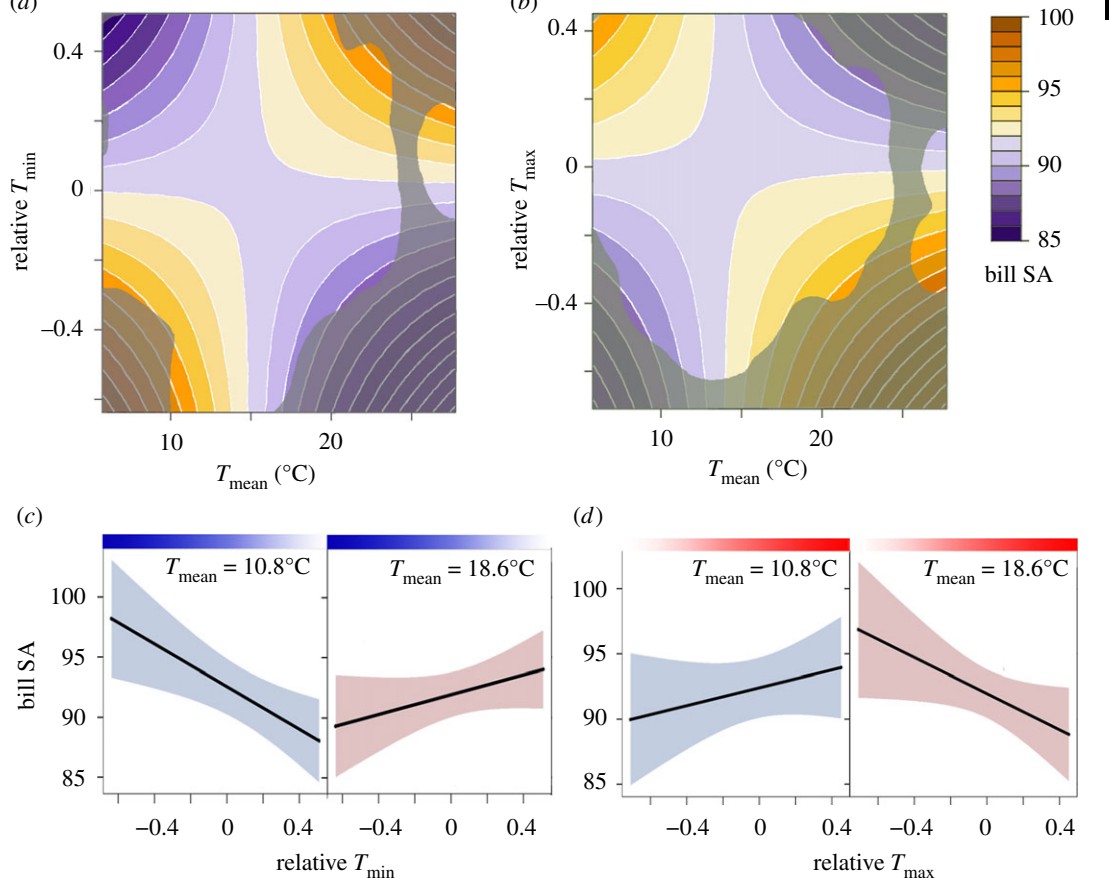

**Figure 2.** The relationship of bill surface area (purple-to-orange gradient, in mm$^2$) with relative minimum temperature (*a*) and relative maximum temperature (*b*) varied depending on the 5-year mean temperature, but in opposite directions. Contour plot area outside of the kernel density space containing 99.9% of datapoints is shaded grey. (*c*) Relative minimum temperature (blue gradient, with darker shades indicating more unusually cool temperatures) was positively associated with bill surface area when the mean temperature was high (confidence interval in pink), but negatively associated when the mean temperature was low (confidence interval in blue); the temperatures shown, 10.8 and 18.6°C, are the 10th and 90th percentiles of mean temperature. (*d*) Relative maximum temperature (red gradient, with darker shades indicating more unusually warm temperatures) exhibited the opposite relationship with bill surface area with regards to mean temperature.

these relationships would be modulated by the environmental context, represented by mean temperature and by precipitation. We found support for the interaction of relative temperature extremity with mean temperature and recent precipitation. However, contrary to our predictions, the relationship between bill surface area and relative temperature extremity was negative in some environmental contexts. Bill surface area was positively associated with relative maximum and minimum temperatures only when the measure of temperature extremity opposed that of the overall thermal context: i.e. relative temperature minimum in warm regimes and relative temperature maximum in cool regimes. By contrast, for relative maximum temperature in warm contexts and relative minimum temperature in cool contexts, bill surface area exhibited a negative association. Similarly, although the prediction of a stronger relationship between bill surface area and relative temperature extremity at higher levels of precipitation was supported, this bill-temperature relationship was positive only for relative temperature minimum, while it was negative for relative temperature maximum.

Bill size in populations of an ecologically similar species, the song sparrow *Melospiza melodia*, is best explained by long-term (30-year mean) temperatures during the season of critical thermal stress: high summer temperatures in California [23] and low winter temperatures in eastern North America [24]. Our results appear to indicate the opposite pattern, with the relationship between relative temperature extremity and bill surface area being positive, as predicted by thermoregulatory theory, only under milder absolute temperature contexts. Whereas song sparrow bills were larger when absolute temperatures were hotter in a hot context (summer), junco bills were larger when relative maximum temperatures were hotter only in a cool context, when mean temperatures were low. Similarly, whereas song sparrow bills were

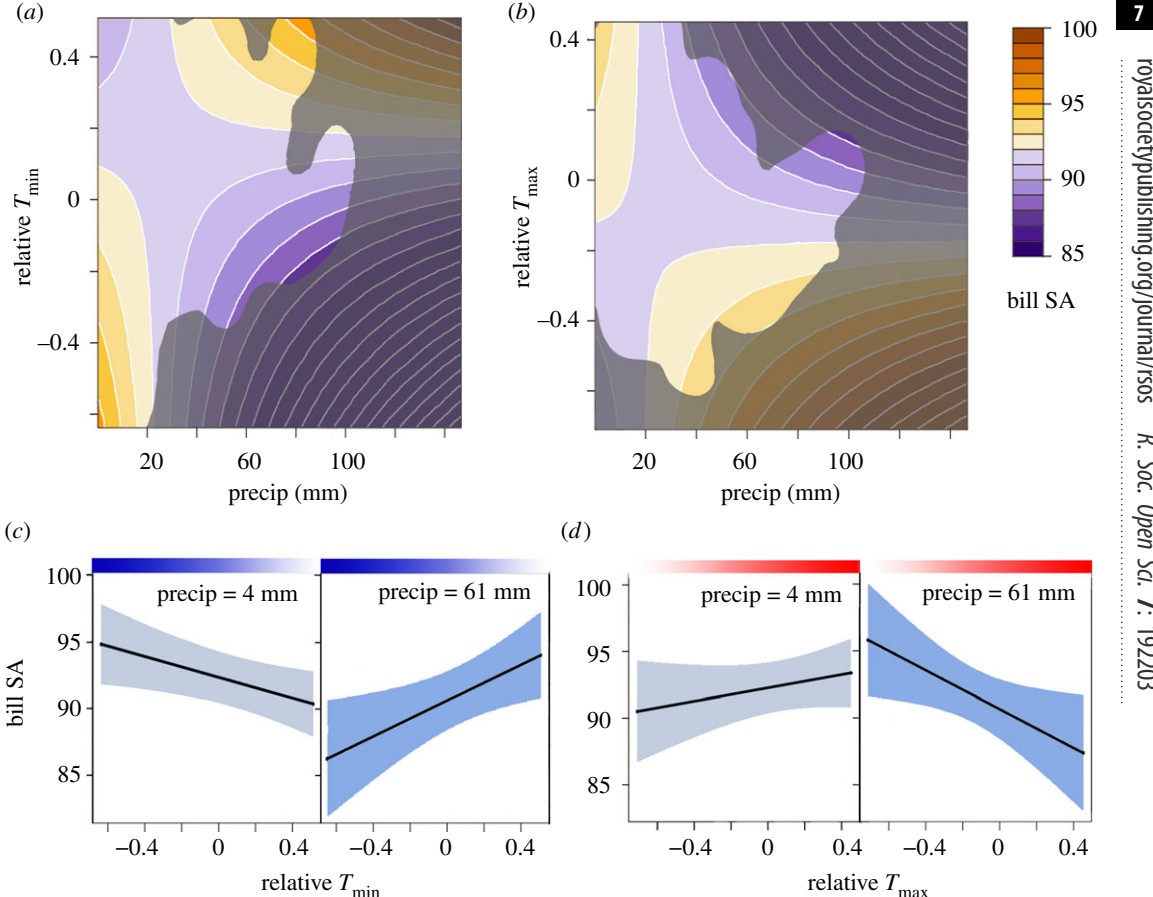

**Figure 3.** The relationship of bill surface area (purple-to-orange gradient, in mm²) with relative minimum temperature (a) and relative maximum temperature (b) varied depending on the previous summer's precipitation, but in opposite directions. Contour plot area outside of the kernel density space containing 99.9% of datapoints is shaded grey. (c) Relative minimum temperature (blue gradient, with darker shades indicating more unusually cool temperatures) was positively associated with bill surface area when precipitation was high (deep blue confidence interval), but there was no relationship when precipitation was low (light blue confidence interval); the precipitation values shown, 4 and 61 mm, are the 10th and 90th percentiles of recent precipitation. (d) By contrast, relative maximum temperature (red gradient, with darker shades indicating more unusually warm temperatures) was negatively associated with bill surface area when precipitation was high, and there was no relationship when precipitation was low.

smaller when absolute temperatures were colder in a cold context (winter), junco bills were smaller when relative minimum temperatures were colder only in a warm context, when mean temperatures were high. Notably, the song sparrow studies differ from this study in examining absolute rather than relative temperature extremes [23,24]. Our results add to a growing appreciation of the effects of temperature on bill morphology by suggesting that while bills may adapt to predictable temperature patterns over long periods as predicted by thermoregulatory theory, bills respond differently to unpredictable short-term extremes.

The unexpected relationships between bill surface area and relative temperature extremity suggests that other selective pressures on bill morphology outweigh thermoregulatory concerns in some cases. A mismatch between bill surface area and thermoregulatory requirements carries the energetic and time costs of metabolic and behavioural compensation [33,34,39], and in extreme cases, an increased risk of death [5]. To overcome these costs, the opposing selective pressure must be similarly strong. In high temperatures, the utility of large bill surface area in thermoregulation is limited because passive heat dissipation is only possible when the environment is cooler than the bird's body; therefore, the selective pressure for larger bills in hot environments is likely reduced, or possibly even reversed to favour smaller heat exchange surfaces [23,29]. Greater humidity can amplify this effect at high temperatures [29], which may explain the lack of the predicted positive relationship between bill surface area and relative temperature minimum at high levels of precipitation. This is

at best an incomplete explanation, however, as such effects should not arise until ambient temperatures approach junco body temperature [23], a circumstance which applies to very few of our specimens.

No similar limit on the thermoregulatory benefits of reduced heat exchange exists at cold temperatures. Further analysis (see electronic supplementary material) showed no support for Bergmann's Rule driving the pattern of bill surface area variation at cold mean temperatures. This could in part be due to the fact that the climate data we used is not sufficiently high-resolution to capture microclimate heterogeneity. Still, our models suggest that some other strong selective pressure is likely relevant in driving the negative relationship between relative minimum temperature and bill surface area. Food acquisition and processing could drive selection on bill size, particularly since the juncos' prey items, seeds and arthropods, are also likely to be impacted by environmental extremes [8,11,15,40]. Although bill surface area is not directly linked to food processing, the linear bill measurements that comprise surface area do have direct influences on food processing function, with larger bill depth and width predicting stronger bite force [41]. The specimens in this study that experienced the coldest 10% of mean temperatures were largely (98%) from mountain habitats (mean ± s.d. elevation: $2672 \pm 693$ m). Seeds comprise a considerable proportion of the junco diet in the California mountains (30% by volume in the summer [42]), and this proportion probably increases during cold snaps, which reduce arthropod prey availability [43]. As high elevations are particularly prone to extreme temperature fluctuations in general [44], larger bills—better suited to processing seeds [41]—may be required to avoid starvation, despite the thermoregulatory cost. This underscores the importance of biotic interactions in mediating responses to abiotic change: particularly in extreme conditions, ecological principles such as Allen's Rule [24] may be overturned by the complexities of ecological interactions.

An ecological interaction which we were not able to examine was the effect of land use change on bill morphology. Bill morphology in the Northern cardinal (*Cardinalis cardinalis*) is associated in various directions with urbanization in some cities [45]; it is possible that similar effects may be present in our dataset. However, because our specimens originate from a variety of locations, including areas that underwent change (e.g. the San Francisco Bay Area) and those that did not (e.g. Yosemite National Park), and because our analysis accounts for effects of year and spatial autocorrelation, any effects of land use change should present as noise reducing our observed relationships, rather than generating spurious relationships.

Whether the relationships we document between bill surface and relative temperature extremity are the result of plasticity or evolution is a key, albeit difficult, question. The time period examined (1 to 5 years) could encompass no more than four complete generations of juncos. Although evolution can occur over short-time periods in cases of extreme selective mortality (e.g. [15,46]), given the lack of noted mass mortality events in this system, plasticity seems most likely. Yet as the capacity for plasticity is itself genetically based, implicating plasticity does not remove us from the realm of genetic change. The capacity for both plasticity and evolutionary change will contribute to species' survival in variable environments [47].

Our results add to growing evidence that the effects of environmental extremes are not simply linear extensions of the effects of more moderate environmental variation and can be qualitatively different [2,11,48]. This underscores the importance of explicitly studying the impacts of thermal extremes as well as means, particularly as extreme events are predicted to increase in frequency due to climate change [1]. Furthermore, thermal extremes may have non-additive affects when combined with other environmental changes (e.g. changes in phenology of prey items [8] or parasites [49]; drought [5]; heavy rain [50]), many of which are also expected to be common as climate change progresses [40].

Data accessibility. The R code and data from this manuscript are available at https://doi.org/10.5061/dryad.8gtht76jh] [51].

Authors' contributions. K.L., K.J.M and K.R.R.H. collected the data. K.L. and T.T.H. analysed the data. All authors wrote and edited the manuscript and approved the final version.

Competing interests. We declare we have no competing interests.

Funding. A Grand Challenges in Biology Postdoctoral Fellowship from the University of Minnesota to K.L. and an NSF Postdoctoral Fellowship to T.T.H.

Acknowledgements. We thank Matthew Symonds and anonymous reviewers for their comments on an earlier version of this manuscript. We thank Carla Cicero, Rauri Bowie, Moe Flannery and the California Academy of Sciences, Eileen Lacey, Steve Beissinger, Doug Renwick, Trent Pingenot, HostGIS, Quintin Stedman and PRISM for their aid with this project.

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
