## [Reviewer comments · Royal Society Open Science]

Review History

RSOS-192203.R0 (Original submission)

Review form: Reviewer 1 (Matthew Symonds)

Is the manuscript scientifically sound in its present form?

Yes

Are the interpretations and conclusions justified by the results?

Yes

Is the language acceptable?

Yes

Do you have any ethical concerns with this paper?

No

Have you any concerns about statistical analyses in this paper?

No

Recommendation?

Accept with minor revision (please list in comments)

Comments to the Author(s)

The authors have done a very good job of meticulously revising this manuscript, and I think it's a very solid piece of work. Of particular improvement is the description of what they mean by temperature extremes, being a RELATIVE term (relative to the usual climate that the bird has experienced) - rather than an absolute judgement (which is what I think of when people discuss temperature extremes). So, in a sense we are talking about unusual temperatures (unusual for the climate) rather than actual biologically 'damaging' extremes for most organisms (like 45 degrees C, or -40 degrees C). This really wasn't clear last time and I think there's still a couple of points in the manuscript where the wording is a little ambiguous (see below) - but overall it's much clearer what exactly they mean. The new analyses including interactions with precipitation adds to the story considerably and the inclusion also of much more supplementary material covers alternative analyses (using absolute temperatures for example) is very good. So I do sense the authors have been stringent into trying to 'cover their bases'. The approach used here of investigating the role of relative extremes of climate is unusual, and I'm not sure is 100% easy to interpret biologically (something the authors are very open about) - but I agree the result is interesting and should stimulate further analysis,

My remaining comments/suggestions are minor:

Title: the use of the term "temperature extremes" in the title with a slightly ambiguous qualification ("context-dependent") still made me think of absolute extremes. I've been trying to think of an alternative way of expressing this to resolve this ambiguity. Perhaps insert 'unusual' before temperature extremes - or 'atypical'?

line 22: insert 'that animals experience' after 'responses to extremes' (again to resolve some ambiguity)

Lines 32-33: (again just for clarity and only a suggestion) instead of 'relative minimum temperature' and 'relative maximum temperature' say 'atypically cold minimum temperature' and 'atypically hot maximum temperature'

Line 70: I would insert a reference to (and brief description of) Allen's rule here - as it's raised later but not actually defined.

Lines 74-91: I still feel a little more explanation / justification is needed for why you are looking at relative extreme temperatures - most of the prior discussion before this paragraph has concerned the effect of direct temperature - not extremes from the norm / variability.

Likewise lines 83-86: While I agree with the prediction here - still it is not clear a priori why you are looking at relative temperature extremity and not just absolute temperature extremity.

Line 139 - because I disagree with what is instantly meant by extreme temperatures I would slightly reword the beginning of this sentence to say 'In this study, "extreme" temperatures are defined in terms of comparison to an established non-extreme baseline'.

Line 149 - I was curious that minimum summer temperatures were used and not minimum winter temperatures. Why? Given other studies (e.g. studies by Danner, Greenberg, Friedman et al's.) it would seem that minimum winter temperature would be more important to consider. Wouldn't you want to compare to the actual colder temperatures the bird experiences during the year.

Line 179: I know that this was a request from another reviewer, but I disagree with the use of tarsus size as the proxy for body size (for exactly the reasons the authors state in the supplementary material). I would have used wing size. However, since the authors have clearly presented the alternative analysis with wing length in the supp material, I don't think this needs to be changed now (although I'd support the authors if they decided to change it back again!).

Line 182 is slightly repetitive of line 174

Lines 235-239: you need to insert a caveat here that the song sparrow study considered absolute not relative extremes (it might be worth teasing this apart this difference in results due to the different measures used). But as stated, this bit of discussion is comparing apples and oranges here, from an analytical point of view.

References: quite a number of references (e.g. 25, 27-30) are missing the journal name.

I look forward to seeing the study published!

Matthew Symonds

Review form: Reviewer 2

Is the manuscript scientifically sound in its present form?

Yes

Are the interpretations and conclusions justified by the results?

Yes

Is the language acceptable?

Yes

Do you have any ethical concerns with this paper?

No

Have you any concerns about statistical analyses in this paper?

Yes

Recommendation?

Major revision is needed (please make suggestions in comments)

Comments to the Author(s)

This is a well-written manuscript that adds to a growing body of literature examining morphological changes in response to climate changes. The novelty of this paper is that the observed morphological changes seem to exhibit a strong relationship to relative temperature deviations (extremity) over a relatively short time span (5 years). Interestingly, the authors report that bill surface area was smaller during times of relative temperature extremity, but only in regions where absolute temperatures opposed the relative climate extremes. While this is indeed an interesting and novel finding, I am not convinced it isn't simply the result of mathematical coupling in the statistical model, leading to inferential bias in the coefficients (please see detailed comments below). I have tried to offer some advice on how the authors could potentially address this concern but also recognize it is not a trivial undertaking.

Line-specific comments:

Abstract

L 26: I think the "novelty" of this measure is overstated. From what I can tell, the measure of short-term relative temperature extremity is a standardized temperature deviation that is commonly used in climate change analyses.

Introduction

The introduction was very well-written – I have no comments to add.

Methods

L 122: I appreciate the justification for using a radius of 15 km to calculate average temperature. However, knowing a bit about the study region, it seems to me that 15 km would average over a lot of the elevation gradient present within the region, which can have a strong influence on the local temperature.

L125: Should be a “.” After “(see below)”

L125:128: Was the mean and standard deviation calculated for all three temperature variables under consideration? Parts of this paragraph seem to conflict with the paragraph above. For example, in line 127 the authors state “...monthly mean temperatures (°C) for the summer months...” When above, the authors state: “... for each calendar month during the five years prior...” If the mean temperatures in this section are referring to calculations that feed into the relative extremity measure, I think that needs to be made explicitly clear.

L 128: Why was only recent precipitation taken into consideration and not the five-year averages in precipitation to be consistent with temperature?

L 148: Why take the mean monthly minimum summer temperature in calculations of relative minimum temperature? Danner and colleagues found the “critical season” for bill size variation was winter. Therefore, wouldn’t it make more sense to use the mean monthly (or absolute) minimum winter temperature in calculations of relative minimum temperature? Presumably these birds were experiencing temperatures throughout the year and not just during the summer/breeding season?

L 167:169: I appreciate wanting to know whether the impacts of relative extreme temperatures (a temporal measure) depend on the absolute temperature context (a spatial measure). However, my concern is that even though relative extreme temperature and mean temperature are uncorrelated with one another, the two measures are still mathematically coupled which could lead to biased conclusions.

For example, the measure for relative minimum temperature is:

$$R_{min} = (([T_{min}]_{-1} - (\sum_{i=1}^5 [T_{min}]_i) / 5) / \sigma_{5yr T_{min}}$$

The interaction between R_{min} and T_{mean} is therefore:

$$(([T_{min}]_{-1} - (\sum_{i=1}^5 [T_{min}]_i) / 5) / \sigma_{5yr T_{min}} \times (\sum_{i=1}^5 [T_{mean}]_i) / 5$$

Because T_{min} and T_{mean} are highly correlated with one another (and therefore substitutable), you essentially have mathematical coupling of variables (even though they are uncorrelated), which will lead to biased coefficient estimates. This interaction is essentially a variation of a quadratic effect which is apparent in Figure 2.

One solution could be – instead of including all terms in a single model – to run one model with relative temperature (i.e., addressing the question: is bill size higher/lower in more extreme years? Suggesting morphological plasticity relates to short-term variation in climate extremes). Then, a second model could be fit using something like relative temperature asymmetry for all specimen (site) pairs along the full temperature gradient (See, for example, Buckley et al. 2013. Ectotherm thermal stress and specialization across altitude and latitude). Therefore, this analysis (including specimen ID as an additional random effect) would address the question: How much greater is the effect of relative T_{max} at a warmer site relative to a colder site? And vice versa for relative T_{min} (i.e., does bill size exhibit a stronger association with temperature in extreme years in more “extreme” locations?). This is the question the interaction term in the original analysis is

trying to address; however, due to the issue with mathematical coupling described above, I'm not convinced by the results.

I have a similar comment regarding the implicit inclusion of temperature variability (in relative temperature extremity) and the explicit inclusion of temperature variability in the current model.

Results

L191-192: The authors should still plot residuals across space and formally test that spatial autocorrelation has in fact been accounted for.

L192: please report (in SI is fine) residual variance/SD for random year effect. This way, the reader can determine whether bill size variation within a given year (e.g., spatial variation) is higher or lower than among years (temporal variation).

L198-200: I think this interaction is overstated for reason's detailed above.

L205: Could the finding that temperature SD had no effect on bill surface area be due to the fact that temperature SD is already implicitly being accounted for in the relative temperature extreme covariate? Do other model coefficients change in sign or magnitude when temperature SD is removed from the model?

L207-211: Could you please plot in SI, the relationship between relative temperatures and elevation? Is there any pattern in the residuals related to elevation? Also, please plot relative temperature vs. year - is relative temperature greater in the latter part of the timeseries? Boxplots of residuals for each year in the timeseries should also be centered around zero and show no patterns of heterogeneity.

Discussion

L256-257: I think the authors should recognize here that the coarse-grained climate datasets used in analyses do not capture fine-grained microclimate heterogeneity that could also lead to the lack of the expected relationship.

L 267-268: If the coldest 10% of mean temps were from high elevations and high elevations also had more extreme temperatures (as indicated by the authors on lines 271-272), then I wonder how much of this interaction effect is driven by elevation. Are there fewer specimens at higher elevation locations?

Figure 2 caption: It was not immediately clear that the "blue gradient" referenced in the caption was referring to the gradient bar along the top axis of panel (c). I suggest moving the panel labels (a), (b), (c) & (d) to the front of the sentence describing them as the "blue gradient" could be confused with panels (a) and (b).

Figure 2: This is a very nice figure. Some minor suggestions: 1) it would be nice to see the points plotted within the interaction gradient in panels (a) and (b). Based on plotting the raw data provided alongside the manuscript, some corners of this gradient have very little to no data. Therefore, I suggest either plotting the sample points in the 2-D space, or, to more accurately portray the interaction, remove values from the grid used to create panels (a) and (b) that are extend beyond the variation in observed data (e.g., if there are no data for mean temperatures < 10 & relative Tmin > 0.3, those values should be filled with NA instead of being represented by a dark blue color). Also, please plot the raw data in panels (c) and (d) like you have in Figure 3c,d.

Figure 3: Same comments as above with respect to panel labels and only plotting values in the interaction space that are within the bounds of the observed data. Lastly, a very minor point: I'd suggest keeping color scheme the same for Figure 3c,d as for Figure 2c,d (e.g., blue for low precipitation error ribbon; red for high precipitation ribbon).

Decision letter (RSOS-192203.R0)

18-Feb-2020

Dear Dr LaBarbera,

The editors assigned to your paper ("Context-dependent effects of temperature extremes on bill morphology in a songbird") have now received comments from reviewers. We would like you to revise your paper in accordance with the referee and Associate Editor suggestions which can be found below (not including confidential reports to the Editor). Please note this decision does not guarantee eventual acceptance.

Please submit a copy of your revised paper before 12-Mar-2020. Please note that the revision deadline will expire at 00.00am on this date. If we do not hear from you within this time then it will be assumed that the paper has been withdrawn. In exceptional circumstances, extensions may be possible if agreed with the Editorial Office in advance. We do not allow multiple rounds of revision so we urge you to make every effort to fully address all of the comments at this stage. If deemed necessary by the Editors, your manuscript will be sent back to one or more of the original reviewers for assessment. If the original reviewers are not available, we may invite new reviewers.

- Data accessibility

<http://datadryad.org/submit?journalID=RSOS&manu=RSOS-192203>

- **Competing interests**

- **Authors' contributions**

- **Acknowledgements**

- **Funding statement**

on behalf of Professor Emily Standen (Associate Editor) and Kevin Padian (Subject Editor)
openscience@royalsociety.org

Associate Editor's comments (Professor Emily Standen):

Dear Dr. LaBarbera,

We have now received your final reviews of your manuscript entitled Context-dependent effects of temperature extremes on bill morphology in a songbird. In general the reviews are quite positive and the reviewers see nice improvement in the clarity of your work.

There are some fairly major concerns from one of the reviewers regarding the construction of your model and they have made some very constructive suggestions as to how you might clarify

some of their concerns. If you are able to address the concerns of these two reviewers we would be happy to consider your manuscript further.

Sincerely,

EMS

Comments to Author:

Reviewers' Comments to Author:

Reviewer: 1

Comments to the Author(s)

The authors have done a very good job of meticulously revising this manuscript, and I think it's a very solid piece of work. Of particular improvement is the description of what they mean by temperature extremes, being a RELATIVE term (relative to the usual climate that the bird has experienced) - rather than an absolute judgement (which is what I think of when people discuss temperature extremes). So, in a sense we are talking about unusual temperatures (unusual for the climate) rather than actual biologically 'damaging' extremes for most organisms (like 45 degrees C, or -40 degrees C). This really wasn't clear last time and I think there's still a couple of points in the manuscript where the wording is a little ambiguous (see below) - but overall it's much clearer what exactly they mean. The new analyses including interactions with precipitation adds to the story considerably and the inclusion also of much more supplementary material covers alternative analyses (using absolute temperatures for example) is very good. So I do sense the authors have been stringent into trying to 'cover their bases'. The approach used here of investigating the role of relative extremes of climate is unusual, and I'm not sure is 100% easy to interpret biologically (something the authors are very open about) - but I agree the result is interesting and should stimulate further analysis,

My remaining comments/suggestions are minor:

Title: the use of the term "temperature extremes" in the title with a slightly ambiguous qualification ("context-dependent") still made me think of absolute extremes. I've been trying to think of an alternative way of expressing this to resolve this ambiguity. Perhaps insert 'unusual' before temperature extremes - or 'atypical'?

line 22: insert 'that animals experience' after 'responses to extremes' (again to resolve some ambiguity)

Lines 32-33: (again just for clarity and only a suggestion) instead of 'relative minimum temperature' and 'relative maximum temperature' say 'atypically cold minimum temperature' and 'atypically hot maximum temperature'

Line 70: I would insert a reference to (and brief description of) Allen's rule here - as it's raised later but not actually defined.

Lines 74-91: I still feel a little more explanation / justification is needed for why you are looking at relative extreme temperatures - most of the prior discussion before this paragraph has concerned the effect of direct temperature - not extremes from the norm / variability.

Likewise lines 83-86: While I agree with the prediction here - still it is not clear a priori why you are looking at relative temperature extremity and not just absolute temperature extremity.

Line 139 - because I disagree with what is instantly meant by extreme temperatures I would slightly reword the beginning of this sentence to say 'In this study, "extreme" temperatures are defined in terms of comparison to an established non-extreme baseline'.

Line 149 - I was curious that minimum summer temperatures were used and not minimum winter temperatures. Why? Given other studies (e.g. studies by Danner, Greenberg, Friedman et al's.) it would seem that minimum winter temperature would be more important to consider. Wouldn't you want to compare to the actual colder temperatures the bird experiences during the year.

Line 179: I know that this was a request from another reviewer, but I disagree with the use of tarsus size as the proxy for body size (for exactly the reasons the authors state in the supplementary material). I would have used wing size. However, since the authors have clearly presented the alternative analysis with wing length in the supp material, I don't think this needs to be changed now (although I'd support the authors if they decided to change it back again!).

Line 182 is slightly repetitive of line 174

Lines 235-239: you need to insert a caveat here that the song sparrow study considered absolute not relative extremes (it might be worth teasing this apart this difference in results due to the different measures used). But as stated, this bit of discussion is comparing apples and oranges here, from an analytical point of view.

References: quite a number of references (e.g. 25, 27-30) are missing the journal name.

I look forward to seeing the study published!

Matthew Symonds

Reviewer: 2

Comments to the Author(s)

This is a well-written manuscript that adds to a growing body of literature examining morphological changes in response to climate changes. The novelty of this paper is that the observed morphological changes seem to exhibit a strong relationship to relative temperature deviations (extremity) over a relatively short time span (5 years). Interestingly, the authors report that bill surface area was smaller during times of relative temperature extremity, but only in regions where absolute temperatures opposed the relative climate extremes. While this is indeed an interesting and novel finding, I am not convinced it isn't simply the result of mathematical coupling in the statistical model, leading to inferential bias in the coefficients (please see detailed comments below). I have tried to offer some advice on how the authors could potentially address this concern but also recognize it is not a trivial undertaking.

Line-specific comments:

Abstract

L 26: I think the "novelty" of this measure is overstated. From what I can tell, the measure of short-term relative temperature extremity is a standardized temperature deviation that is commonly used in climate change analyses.

Introduction

The introduction was very well-written – I have no comments to add.

Methods

L 122: I appreciate the justification for using a radius of 15 km to calculate average temperature. However, knowing a bit about the study region, it seems to me that 15 km would average over a

lot of the elevation gradient present within the region, which can have a strong influence on the local temperature.

L125: Should be a “.” After “(see below)”

L125:128: Was the mean and standard deviation calculated for all three temperature variables under consideration? Parts of this paragraph seem to conflict with the paragraph above. For example, in line 127 the authors state “...monthly mean temperatures (°C) for the summer months...” When above, the authors state: “... for each calendar month during the five years prior...” If the mean temperatures in this section are referring to calculations that feed into the relative extremity measure, I think that needs to be made explicitly clear.

L 128: Why was only recent precipitation taken into consideration and not the five-year averages in precipitation to be consistent with temperature?

L 148: Why take the mean monthly minimum summer temperature in calculations of relative minimum temperature? Danner and colleagues found the “critical season” for bill size variation was winter. Therefore, wouldn’t it make more sense to use the mean monthly (or absolute) minimum winter temperature in calculations of relative minimum temperature? Presumably these birds were experiencing temperatures throughout the year and not just during the summer/breeding season?

L 167:169: I appreciate wanting to know whether the impacts of relative extreme temperatures (a temporal measure) depend on the absolute temperature context (a spatial measure). However, my concern is that even though relative extreme temperature and mean temperature are uncorrelated with one another, the two measures are still mathematically coupled which could lead to biased conclusions.

For example, the measure for relative minimum temperature is:

$$R_{min} = (([T_{min}]_{-1} - (\sum_{i=1}^5 [T_{min}]_i) / 5) / \sigma_{5yr T_{min}}$$

The interaction between R_{min} and T_{mean} is therefore:

$$(([T_{min}]_{-1} - (\sum_{i=1}^5 [T_{min}]_i) / 5) / \sigma_{5yr T_{min}} \times (\sum_{i=1}^5 [T_{mean}]_i) / 5$$

Because T_{min} and T_{mean} are highly correlated with one another (and therefore substitutable), you essentially have mathematical coupling of variables (even though they are uncorrelated), which will lead to biased coefficient estimates. This interaction is essentially a variation of a quadratic effect which is apparent in Figure 2.

One solution could be – instead of including all terms in a single model – to run one model with relative temperature (i.e., addressing the question: is bill size higher/lower in more extreme years? Suggesting morphological plasticity relates to short-term variation in climate extremes). Then, a second model could be fit using something like relative temperature asymmetry for all specimen (site) pairs along the full temperature gradient (See, for example, Buckley et al. 2013. Ectotherm thermal stress and specialization across altitude and latitude). Therefore, this analysis (including specimen ID as an additional random effect) would address the question: How much greater is the effect of relative T_{max} at a warmer site relative to a colder site? And vice versa for relative T_{min} (i.e., does bill size exhibit a stronger association with temperature in extreme years in more “extreme” locations?). This is the question the interaction term in the original analysis is trying to address; however, due to the issue with mathematical coupling described above, I’m not convinced by the results.

I have a similar comment regarding the implicit inclusion of temperature variability (in relative temperature extremity) and the explicit inclusion of temperature variability in the current model.

Results

L191-192: The authors should still plot residuals across space and formally test that spatial autocorrelation has in fact been accounted for.

L192: please report (in SI is fine) residual variance/SD for random year effect. This way, the reader can determine whether bill size variation within a given year (e.g., spatial variation) is higher or lower than among years (temporal variation).

L198-200: I think this interaction is overstated for reason's detailed above.

L205: Could the finding that temperature SD had no effect on bill surface area be due to the fact that temperature SD is already implicitly being accounted for in the relative temperature extreme covariate? Do other model coefficients change in sign or magnitude when temperature SD is removed from the model?

L207-211: Could you please plot in SI, the relationship between relative temperatures and elevation? Is there any pattern in the residuals related to elevation? Also, please plot relative temperature vs. year - is relative temperature greater in the latter part of the timeseries? Boxplots of residuals for each year in the timeseries should also be centered around zero and show no patterns of heterogeneity.

Discussion

L256-257: I think the authors should recognize here that the coarse-grained climate datasets used in analyses do not capture fine-grained microclimate heterogeneity that could also lead to the lack of the expected relationship.

L 267-268: If the coldest 10% of mean temps were from high elevations and high elevations also had more extreme temperatures (as indicated by the authors on lines 271-272), then I wonder how much of this interaction effect is driven by elevation. Are there fewer specimens at higher elevation locations?

Figure 2 caption: It was not immediately clear that the "blue gradient" referenced in the caption was referring to the gradient bar along the top axis of panel (c). I suggest moving the panel labels (a), (b), (c) & (d) to the front of the sentence describing them as the "blue gradient" could be confused with panels (a) and (b).

Figure 2: This is a very nice figure. Some minor suggestions: 1) it would be nice to see the points plotted within the interaction gradient in panels (a) and (b). Based on plotting the raw data provided alongside the manuscript, some corners of this gradient have very little to no data. Therefore, I suggest either plotting the sample points in the 2-D space, or, to more accurately portray the interaction, remove values from the grid used to create panels (a) and (b) that are extend beyond the variation in observed data (e.g., if there are no data for mean temperatures < 10 & relative Tmin > 0.3, those values should be filled with NA instead of being represented by a dark blue color). Also, please plot the raw data in panels (c) and (d) like you have in Figure 3c,d.

Figure 3: Same comments as above with respect to panel labels and only plotting values in the interaction space that are within the bounds of the observed data. Lastly, a very minor point: I'd suggest keeping color scheme the same for Figure 3c,d as for Figure 2c,d (e.g., blue for low precipitation error ribbon; red for high precipitation ribbon).

Author's Response to Decision Letter for (RSOS-192203.R0)

See Appendix A.

Decision letter (RSOS-192203.R1)

10-Mar-2020

Dear Dr LaBarbera,

It is a pleasure to accept your manuscript entitled "Context-dependent effects of relative temperature extremes on bill morphology in a songbird" in its current form for publication in Royal Society Open Science.

on behalf of Professor Emily Standen (Associate Editor) and Kevin Padian (Subject Editor)
openscience@royalsociety.org

Associate Editor Comments to Author (Professor Emily Standen):

Associate Editor

Comments to the Author:

Dear Dr. LaBarera,

I would like to take this opportunity to commend your thorough and clear address to the most recent reviewer comments on your manuscript entitled Context-dependent effects of relative temperature extremes on bill morphology in a songbird.

I have returned your well formed comments to the reviewer who had requested them and have asked for a quick confirmation response for how you have addressed their concerns.

Hopefully this will be a very quick turn around.

Sincerely,
EMS

Appendix A

2 March 2020

Dear Dr. Standen,

Thank you very much for your comments on our recently submitted article. Enclosed is our revised manuscript (#RSOS-192203), "Context-dependent effects of relative temperature extremes on bill morphology in a songbird." We would like to thank both reviewers and you for your feedback. We have addressed all of the reviewer comments and believe they have resulted in significant improvements to the manuscript. We have explicitly checked for issues of "mathematical coupling" as raised by Reviewer 2 and confirm that they are not driving our results. We hope you will consider the revised manuscript for publication in Royal Society Open Science.

This work is original research that has not been published or submitted elsewhere. All authors have read and approved the manuscript. Thank you for your consideration.

Associate Editor's comments (Professor Emily Standen):

Dear Dr. LaBarbera,

We have now received your final reviews of your manuscript entitled Context-dependent effects of temperature extremes on bill morphology in a songbird. In general the reviews are quite positive and the reviewers see nice improvement in the clarity of your work.

There are some fairly major concerns from one of the reviewers regarding the construction of your model and they have made some very constructive suggestions as to how you might clarify some of their concerns. If you are able to address the concerns of these two reviewers we would be happy to consider your manuscript further.

**Sincerely,
EMS**

Thank you for handling our manuscript. We have responded to all comments below and hope that the manuscript is now acceptable for publication.

Comments to Author:

Reviewers' Comments to Author:

Reviewer: 1

Comments to the Author(s)

The authors have done a very good job of meticulously revising this manuscript, and I think it's a very solid piece of work. Of particular improvement is the description of what

they mean by temperature extremes, being a **RELATIVE** term (relative to the usual climate that the bird has experienced) - rather than an absolute judgement (which is what I think of when people discuss temperature extremes). So, in a sense we are talking about unusual temperatures (unusual for the climate) rather than actual biologically 'damaging' extremes for most organisms (like 45 degrees C, or -40 degrees C). This really wasn't clear last time and I think there's still a couple of points in the manuscript where the wording is a little ambiguous (see below) - but overall it's much clearer what exactly they mean. The new analyses including interactions with precipitation adds to the story considerably and the inclusion also of much more supplementary material covers alternative analyses (using absolute temperatures for example) is very good. So I do sense the authors have been stringent into trying to 'cover their bases'. The approach used here of investigating the role of relative extremes of climate is unusual, and I'm not sure is 100% easy to interpret biologically (something the authors are very open about) - but I agree the result is interesting and should stimulate further analysis,

Thank you for your positive feedback about our revision.

My remaining comments/suggestions are minor:

Title: the use of the term "temperature extremes" in the title with a slightly ambiguous qualification ("context-dependent") still made me think of absolute extremes. I've been trying to think of an alternative way of expressing this to resolve this ambiguity. Perhaps insert 'unusual' before temperature extremes - or 'atypical'?

We have inserted "relative" as a qualifier, since that is the term used elsewhere in the paper.

line 22: insert 'that animals experience' after 'responses to extremes' (again to resolve some ambiguity)

Unfortunately we are tight up against the 200-word abstract maximum, and can't find a way to add these words without removing words that create worse ambiguity elsewhere.

Lines 32-33: (again just for clarity and only a suggestion) instead of 'relative minimum temperature' and 'relative maximum temperature' say 'atypically cold minimum temperature' and 'atypically hot maximum temperature'

We have made the suggested change (lines 32-33).

Line 70: I would insert a reference to (and brief description of) Allen's rule here - as it's raised later but not actually defined.

Thank you for pointing out this omission; we have added a brief description and references at line 65-66.

Lines 74-91: I still feel a little more explanation / justification is needed for why you are looking at relative extreme temperatures - most of the prior discussion before this paragraph has concerned the effect of direct temperature - not extremes from the norm / variability.

We have added several sentences to hopefully clarify this further (lines 75-80), including a reference to Janzen 1967.

Likewise lines 83-86: While I agree with the prediction here - still it is not clear a priori why you are looking at relative temperature extremity and not just absolute temperature extremity.

(see response to the previous comment)

Line 139 - because I disagree with what is instantly meant by extreme temperatures I would slightly reword the beginning of this sentence to say 'In this study, "extreme" temperatures are defined in terms of comparison to an established non-extreme baseline'.

We have made the suggested change (line 140).

Line 149 - I was curious that minimum summer temperatures were used and not minimum winter temperatures. Why? Given other studies (e.g. studies by Danner, Greenberg, Friedman et al's.) it would seem that minimum winter temperature would be more important to consider. Wouldn't you want to compare to the actual colder temperatures the bird experiences during the year.

The juncos generally migrate for the winter, not nearly as far as long-distance migratory populations but still an adequate distance to render measurements of the climate at their breeding location irrelevant to what they are experiencing in the winter. Since we don't know where they were during the winter, we can't obtain climate data for those months. We have added this explanation at lines 124-129.

Line 179: I know that this was a request from another reviewer, but I disagree with the use of tarsus size as the proxy for body size (for exactly the reasons the authors state in the supplementary material). I would have used wing size. However, since the authors have clearly presented the alternative analysis with wing length in the supp material, I don't think this needs to be changed now (although I'd support the authors if they decided to change it back again!).

We will leave the analyses as is for now, as it doesn't change the results to use tarsus vs. wing chord, and will defer to the editor about which body size proxy to include in the final manuscript.

Line 182 is slightly repetitive of line 174

We have made this sentence more concise, but feel that including some of this information may be pertinent for readers who are familiar with GLMMs but not with GAMMs (line 180-181).

Lines 235-239: you need to insert a caveat here that the song sparrow study considered absolute not relative extremes (it might be worth teasing this apart this difference in results due to the different measures used). But as stated, this bit of discussion is comparing apples and oranges here, from an analytical point of view.

We have added a caveat that the song sparrow study considers absolute rather than relative extremes (line 251-252)

References: quite a number of references (e.g. 25, 27-30) are missing the journal name.

Apologies for this oversight. All references should be appropriately formatted now.

I look forward to seeing the study published!

Matthew Symonds

Thank you for your constructive feedback, which has strengthened the manuscript!

Reviewer: 2

Comments to the Author(s)

This is a well-written manuscript that adds to a growing body of literature examining morphological changes in response to climate changes. The novelty of this paper is that the observed morphological changes seem to exhibit a strong relationship to relative temperature deviations (extremity) over a relatively short time span (5 years). Interestingly, the authors report that bill surface area was smaller during times of relative temperature extremity, but only in regions where absolute temperatures opposed the relative climate extremes. While this is indeed an interesting and novel finding, I am not convinced it isn't simply the result of mathematical coupling in the statistical model, leading to inferential bias in the coefficients (please see detailed comments below). I have tried to offer some advice on how the authors could potentially address this concern but also recognize it is not a trivial undertaking.

Thank you for your feedback. We appreciate the mathematical concerns you bring up. We have explored the issue by constructing additional models and hope that this will alleviate any concerns about potential coupling/covariance in the models. Please see below for a more detailed explanation.

Line-specific comments:

Abstract

L 26: I think the “novelty” of this measure is overstated. From what I can tell, the measure of short-term relative temperature extremity is a standardized temperature deviation that is commonly used in climate change analyses.

We have deleted the word “novel” (line 27).

Introduction

The introduction was very well-written – I have no comments to add.

Thank you for the positive feedback!

Methods

L 122: I appreciate the justification for using a radius of 15 km to calculate average temperature. However, knowing a bit about the study region, it seems to me that 15 km would average over a lot of the elevation gradient present within the region, which can have a strong influence on the local temperature.

We believe 15 km is a reasonable choice for buffer size, given that there is a trade-off here, since including too small of a buffer could risk misrepresenting the range of temperatures potentially experienced by a given specimen.

L125: Should be a “.” After “(see below)”

Done

L125:128: Was the mean and standard deviation calculated for all three temperature variables under consideration? Parts of this paragraph seem to conflict with the paragraph above. For example, in line 127 the authors state “...monthly mean temperatures (°C) for the summer months...” When above, the authors state: “... for each calendar month during the five years prior...” If the mean temperatures in this section are referring to calculations that feed into the relative extremity measure, I think that needs to be made explicitly clear.

We have edited line 121 to clarify that in all cases we were only using the months April through August. We have added the statement that these values were then used for calculation of the variables (lines 123-124).

L 128: Why was only recent precipitation taken into consideration and not the five-year averages in precipitation to be consistent with temperature?

Because precipitation is functioning differently than mean temperature in the model: the five-year temperature mean represents the overall, longer-term temperature context, while precipitation represents the humidity present during the short-term relative temperature extremes. We don't expect precipitation to be interacting with relative extremes as an overall regime, but rather as an immediate modifier of the extreme (because humidity affects heat dissipation). We have added this explanation to the Methods (lines 135-138).

L 148: Why take the mean monthly minimum summer temperature in calculations of relative minimum temperature? Danner and colleagues found the “critical season” for bill size variation was winter. Therefore, wouldn't it make more sense to use the mean monthly (or absolute) minimum winter temperature in calculations of relative minimum temperature? Presumably these birds were experiencing temperatures throughout the year and not just during the summer/breeding season?

The juncos generally migrate for the winter, not nearly as far as long-distance migratory populations but still an adequate distance to render measurements of the climate at their breeding location irrelevant to what they are experiencing in the winter. Since we don't know where they were during the winter, we can't obtain climate data for those months. We have added this explanation at lines 124-129.

L 167:169: I appreciate wanting to know whether the impacts of relative extreme temperatures (a temporal measure) depend on the absolute temperature context (a spatial measure). However, my concern is that even though relative extreme temperature and mean temperature are uncorrelated with one another, the two measures are still mathematically coupled which could lead to biased conclusions.

For example, the measure for relative minimum temperature is:

$$R_{min} = (([Tmin]_{-1} - (\sum_{i=1}^5 [Tmin]_{-i})/5) / \sigma_{5yrTmin})$$

The interaction between R_{min} and T_{mean} is therefore:

$$(([Tmin]_{-1} - (\sum_{i=1}^5 [Tmin]_{-i})/5) / \sigma_{5yrTmin}) \times (\sum_{i=1}^5 [Tmean]_{-i})/5$$

Because T_{min} and T_{mean} are highly correlated with one another (and therefore substitutable), you essentially have mathematical coupling of variables (even though they are uncorrelated), which will lead to biased coefficient estimates. This interaction is essentially a variation of a quadratic effect which is apparent in Figure 2.

One solution could be – instead of including all terms in a single model – to run one model with relative temperature (i.e., addressing the question: is bill size higher/lower in more extreme years? Suggesting morphological plasticity relates to short-term variation in climate extremes). Then, a second model could be fit using something like relative temperature asymmetry for all specimen (site) pairs along the full temperature gradient (See, for example, Buckley et al. 2013. Ectotherm thermal stress and specialization across altitude and latitude). Therefore, this analysis (including specimen ID as an additional random effect) would address the question: How much greater is the effect of relative T_{max} at a warmer site relative to a colder site? And vice versa for relative T_{min} (i.e., does bill size exhibit a stronger association with temperature in extreme years in more “extreme” locations?). This is the question the interaction term in the original analysis is trying to address; however, due to the issue with mathematical coupling described above, I’m not convinced by the results.

This is an important concern to check and we thank the reviewer for pointing it out. We are grateful for the suggestion of analysis, but unfortunately that particular approach is not feasible: our study does not have a paired design, as is required by the analysis, and testing all possible pairs would lead to extreme pseudoreplication, which could not be accounted for using random effects as suggested because a random effect of ID would mean a random effect with 522 unranked categories, which is not feasible for a dataset of this size.

The lack of correlation between our variables is a moderate reassurance, since mathematical coupling would be expected to result in correlation. Mathematical coupling that did not result in linear correlation would also be unlikely to impact our results, as strongly nonlinear relationships should not be picked up by the linear model. However, to be certain, we did an additional analysis:

To check whether mathematical coupling could be driving our results, we additionally ran a “worst-case scenario” analysis designed to yield the maximum possible mathematical coupling in the variables of concern. To do this, we explicitly substituted T_{mean} for the T_{min} and T_{max} variables in the relative extreme measure. This produces the maximum possible level of

mathematical coupling between these variables. If mathematical coupling is contributing to our results in the true model, then this model should reproduce those results as strongly or more so. Instead, we found no replication of the climate results (the variable substituting Tmean for Tmin/Tmax is called “dummy_Tmean”):

Formula:

$$\text{bill_sa} \sim \text{X1yr_ppt} * \text{dummy_Tmean} + \text{dummy_Tmean} * \text{X5yr_tmean} + \text{X5yr_tsd} + \text{s(lat, long)} + \text{tarsus} + \text{sex} + \text{thur} + \text{month}$$

Parametric coefficients:

	Estimate	Std. Error	t value	Pr(> t)
(Intercept)	60.28987	9.20473	6.550	1.42e-10 ***
X1yr_ppt	-0.00866	0.01779	-0.487	0.62656
dummy_Tmean	-2.58311	2.60666	-0.991	0.32218
X5yr_tmean	-0.03253	0.11831	-0.275	0.78344
X5yr_tsd	0.258850	0.44156	0.586	0.55799
tarsus	1.677310	0.45072	3.721	0.00022 ***
sexM	1.506830	0.50206	3.001	0.00282 **
thurY	-3.39159	1.03853	-3.266	0.00117 **
month	-0.03396	0.17649	-0.192	0.84749
X1yr_ppt:dummy_Tmean	0.039750	0.10123	0.393	0.69473
dummy_Tmean:X5yr_tmean	0.34202	0.59408	0.576	0.56506

Signif. codes: 0 ‘***’ 0.001 ‘**’ 0.01 ‘*’ 0.05 ‘.’ 0.1 ‘ ’ 1

Approximate significance of smooth terms:

	edf	Ref.df	F	p-value
s(lat,long)	5.422	5.422	3.196	0.0058 **

Signif. codes: 0 ‘***’ 0.001 ‘**’ 0.01 ‘*’ 0.05 ‘.’ 0.1 ‘ ’ 1

Based on these results, we conclude that mathematical coupling between Tmean and the relative extreme measures is not contributing to our results.

I have a similar comment regarding the implicit inclusion of temperature variability (in relative temperature extremity) and the explicit inclusion of temperature variability in the current model.

As above, we substituted Tsd into the relative extreme formula to create a variable (“dummy_Tsd”) which is maximally mathematically coupled with Tsd. This model also failed to reproduce the results of the true model:

Formula:

$$\text{bill_sa} \sim \text{X1yr_ppt} * \text{dummy_Tsd} + \text{dummy_Tsd} * \text{X5yr_tmean} + \text{X5yr_tsd} + \text{s(lat, long)} + \text{tarsus} + \text{sex} + \text{thur} + \text{month}$$

Parametric coefficients:

	Estimate	Std. Error	t value	Pr(> t)
(Intercept)	60.409889	9.241775	6.537	1.54e-10 ***
X1yr_ppt	0.005806	0.019474	0.298	0.765711
dummy_Tsd	-1.497027	3.021325	-0.495	0.620472
X5yr_tmean	-0.029943	0.118025	-0.254	0.799827
X5yr_tsd	0.090732	0.437470	0.207	0.835779
tarsus	1.702174	0.451835	3.767	0.000185 ***
sexM	1.499104	0.501089	2.992	0.002910 **
thurY	-3.500052	1.055868	-3.315	0.000983 ***
month	-0.048006	0.176427	-0.272	0.785656
X1yr_ppt:dummy_Tsd	-0.066844	0.096723	-0.691	0.489826
dummy_Tsd:X5yr_tmean	-0.127489	0.743751	-0.171	0.863967

Signif. codes:	0 '***'	0.001 '**'	0.01 '*'	0.05 '.' 0.1 ' ' 1

Therefore, it is unlikely that mathematical coupling with Tsd is driving our results.

Results

L191-192: The authors should still plot residuals across space and formally test that spatial autocorrelation has in fact been accounted for.

We have tested for spatial autocorrelation in the model residuals using Moran's I and found no significant correlation ($p = 0.1333$). Added this to the paper at line 191-193.

L192: please report (in SI is fine) residual variance/SD for random year effect. This way, the reader can determine whether bill size variation within a given year (e.g., spatial variation) is higher or lower than among years (temporal variation).

Added this to line 194.

L198-200: I think this interaction is overstated for reason's detailed above.

Please see our response to your above comment.

L205: Could the finding that temperature SD had no effect on bill surface area be due to the fact that temperature SD is already implicitly being accounted for in the relative temperature extreme covariate? Do other model coefficients change in sign or magnitude when temperature SD is removed from the model?

Model coefficients do not notably change in sign or magnitude when temperature SD is removed from the model (no changes in sign; only very minor changes in magnitude; see below output table). For this reason, we do not think that temperature SD is driving the patterns we found.

```

Family: gaussian
Link function: identity

Formula:
bill_sa ~ tmin * X5yr_tmean + tmax * X5yr_tmean + tmin * X1yr_ppt +
          tmax * X1yr_ppt + s(lat, long) + tarsus + sex + thur + month

Parametric coefficients:
              Estimate Std. Error t value Pr(>|t|)
(Intercept)  62.53735    8.97806   6.966 1.03e-11 ***
tmin         -5.18295    1.88036  -2.756 0.006057 **
X5yr_tmean   0.06068    0.10655   0.570 0.569244
tmax         4.02716    2.55364   1.577 0.115417
X1yr_ppt     -0.01650    0.01722  -0.958 0.338390
tarsus       1.58302    0.44327   3.571 0.000389 ***
sexM         1.59656    0.49577   3.220 0.001363 **
thurY       -2.97042    1.00574  -2.953 0.003289 **
month        -0.02031    0.16794  -0.121 0.903800
tmin:X5yr_tmean 1.54704    0.51827   2.985 0.002974 **
X5yr_tmean:tmax -1.27558    0.56986  -2.238 0.025631 *
tmin:X1yr_ppt  0.19490    0.06529   2.985 0.002971 **
tmax:X1yr_ppt -0.17840    0.08153  -2.188 0.029120 *
---
Signif. codes:  0 '***' 0.001 '**' 0.01 '*' 0.05 '.' 0.1 ' ' 1

Approximate significance of smooth terms:
              edf Ref.df    F p-value
s(lat,long)  5.412  5.412 2.997 0.00932 **
---
Signif. codes:  0 '***' 0.001 '**' 0.01 '*' 0.05 '.' 0.1 ' ' 1

R-sq.(adj) =  0.136
lmer.REML = 3250.2  Scale est. = 28.947    n = 522
> |

```

L207-211: Could you please plot in SI, the relationship between relative temperatures and elevation? Is there any pattern in the residuals related to elevation?

Also, please plot relative temperature vs. year – is relative temperature greater in the latter part of the timeseries? Boxplots of residuals for each year in the timeseries should also be centered around zero and show no patterns of heterogeneity.

There is no apparent relationship between elevation and minimum or maximum relative temperature (see below, top pair of plots). There is no pattern in the residuals related to elevation (see below, next plot). There is no apparent relationship between relative minimum or maximum temperature and year (see below). There is no apparent heterogeneity in the residuals with relation to year (see below, last plot, series of boxplots); residuals are not perfectly centered around zero in each year, but there is no pattern with this changing as year increases or decreases. There is considerable heterogeneity among years, but no pattern of

increasing/decreasing heterogeneity over time (the heterogeneity is an important reason why we included year as a random effect).

Discussion

L256-257: I think the authors should recognize here that the coarse-grained climate datasets used in analyses do not capture fine-grained microclimate heterogeneity that could also lead to the lack of the expected relationship.

We have added this caveat to the manuscript (lines 262-263).

L 267-268: If the coldest 10% of mean temps were from high elevations and high elevations also had more extreme temperatures (as indicated by the authors on lines 271-272), then I wonder how much of this interaction effect is driven by elevation. Are there fewer specimens at higher elevation locations?

Elevation was correlated with mean temperature and with precipitation values, but in the case of the data examined here it was not correlated with either of the measures of relative temperature extreme (see plots in earlier response). The line about temperature extremes at high elevation referred to a reference, not to our data; we have rephrased to make this more clear. Less than 10% of the specimens are from elevations above 2500m.

Figure 2 caption: It was not immediately clear that the “blue gradient” referenced in the caption was referring to the gradient bar along the top axis of panel (c). I suggest moving the panel labels (a), (b), (c) & (d) to the front of the sentence describing them as the “blue gradient” could be confused with panels (a) and (b).

We have rearranged the caption and believe it should be more clear now (lines 336-344)

Figure 2: This is a very nice figure. Some minor suggestions: 1) it would be nice to see the points plotted within the interaction gradient in panels (a) and (b). Based on plotting the raw data provided alongside the manuscript, some corners of this gradient have very little to no data. Therefore, I suggest either plotting the sample points in the 2-D space, or, to more accurately portray the interaction, remove values from the grid used to create panels (a) and (b) that are extend beyond the variation in observed data (e.g., if there are no data for mean temperatures < 10 & relative Tmin > 0.3 , those values should be filled with NA instead of being represented by a dark blue color). Also, please plot the raw data in panels (c) and (d) like you have in Figure 3c,d.

Panels a and b: here and for figure 3, we have shaded parts of the graphs that fall outside a 99.9% kernel density plot of the data points. (We have not plotted the data points themselves because they make the graph very visually busy.)

Panels c and d: The version you saw did not include any raw data in either figure's c and d panels - in fig3, those were meant to be raindrops to illustrate high vs. low precipitation. Clearly that was a confusing visual, so we have removed those.

Figure 3: Same comments as above with respect to panel labels and only plotting values in the interaction space that are within the bounds of the observed data. Lastly, a very minor point: I'd suggest keeping color scheme the same for Figure 3c,d as for Figure 2c,d (e.g., blue for low precipitation error ribbon; red for high precipitation ribbon).

We have rearranged the caption and believe it should be more clear now (lines 346-355).

Panels c and d: We hesitate to use the same color scheme as in figure 2 for the confidence bounds, as the colors in fig 2 intuitively evoke warm (red) vs cool (blue), whereas red for high precipitation and blue for low precipitation are non-intuitive; however, we agree that some color scheme is necessary, so we have changed to light blue for low precip and deep blue for high precip.